# HyperRouter: Towards Efficient Training and Inference of Sparse Mixture of Experts via HyperNetwork

**Giang Do**[†]**, Khiem Le**[‡]**, Quang Pham**[⋆,✉]**, TrungTin Nguyen**[⊥]**, Thanh-Nam Doan**[△]**,
**Binh T. Nguyen**[♠]**, Chenghao Liu**[♣]**, Savitha Ramasamy**[⋆]**, Xiaoli Li**[⋆]**, Steven Hoi**[◇]

[⋆] Institute for Infocomm Research (I[2]R), A[∗]STAR, Singapore,
[†] University of Tennessee at Chattanooga, [‡] VinUniversity, [△] Independent Researcher,
[⊥] Univ. Grenoble Alpes, Inria, CNRS, Grenoble INP, LJK, 38000 Grenoble, France,
[♠] AISIA Lab, University of Science, Vietnam National University Ho Chi Minh City,
[♣] Salesforce Research Asia, [◇] Singapore Management University
[✉] Correspondence to: phquang@i2r.a-star.edu.sg

## Abstract

By routing input tokens to only a few split experts, Sparse Mixture-of-Experts has enabled efficient training of large language models. Recent findings suggest that fixing the routers can achieve competitive performance by alleviating the collapsing problem, where all experts eventually learn similar representations. However, this strategy has two key limitations: (i) the policy derived from random routers might be suboptimal, and (ii) it requires extensive resources during training and evaluation, leading to limited efficiency gains. This work introduces HyperRouter, which dynamically generates the router's parameters through a fixed hypernetwork and trainable embeddings to achieve a balance between training the routers and freezing them to learn an improved routing policy. Extensive experiments across a wide range of tasks demonstrate the superior performance and efficiency gains of HyperRouter compared to existing routing methods. Our implementation is publicly available at https://github.com/giangdip2410/HyperRouter.

## 1 Introduction

Recent years have witnessed tremendous successes of the Transformer model (Vaswani et al., 2017) and its variants across a wide range of tasks, ranging from natural language and speech processing (Bao et al., 2022b; Gulati et al., 2020), computer vision (Dosovitskiy et al., 2021; Ruiz et al., 2021; Bao et al., 2022a), reinforcement learning (Chow et al., 2023), to life sciences (Rives et al., 2021). Since then, scaling up to larger models has become the prevailing approach for advancing the state-of-the-art in pre-training and finetuning tasks. However, training such large models comes with a high computational cost (Lin et al., 2022); therefore, there is a growing need to develop efficient strategies that facilitate the training of large language models (LLMs) (Fedus et al., 2022a). One

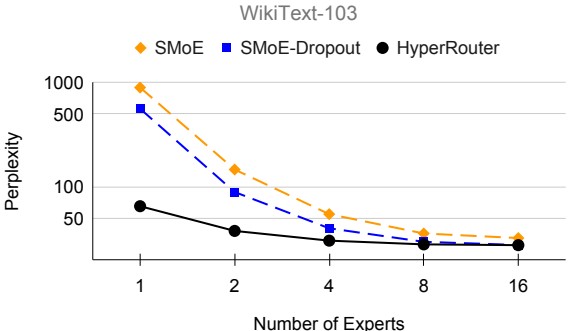

Figure 1: Perplexity (log-scaled) on the WikiText-103 dataset with varying numbers of experts used for inference. All methods have the same FLOPs.

of the most effective strategies thus far is the Sparse Mixture-of-Experts (SMoE) (Shazeer et al., 2017; Fedus et al., 2022b), which utilizes routers to direct each input token to a small subset of network parameters (experts). SMoE improves both efficiency and performance compared to dense training approaches (Lewis et al., 2021; Artetxe et al., 2022; Zhang et al., 2022; Du et al., 2022).

Despite the encouraging results, SMoE has been found to encounter the issue of *representation collapse*, where all experts converge to similar representations (Chi et al., 2022; Chen et al., 2022). This problem arises due to the learning process of the router encouraging experts to cluster around a centroid (Chi et al., 2022). Therefore, significant efforts have been dedicated to addressing the representation collapse issue while preserving the simplicity and efficiency of SMoE training. To this end, one of the most effective strategies is freezing the router, as demonstrated by SMoE-Dropout (Chen et al., 2023), where a randomly initialized router remains fixed throughout the training process. Additionally, Chen et al. (2023) found that progressively increasing the number of selected experts can be beneficial. However, we argue that such a naive strategy exhibits two key limitations. Firstly, the random routing policy may be sub-optimal, which

hinders the overall training process. Secondly, we will show in Section 3.3 that a fixed router will restrict the model's representation capabilities, necessitating a progressive increase in the number of chosen experts to achieve satisfactory performance. Therefore, SMoE-Dropout inherently suffers from limited representation and does not offer efficiency gains during inference.

This work introduces a novel approach called HyperRouter to address the trade-off between fixed and trainable routers in SMoE training. Instead of employing a random but fixed router, HyperRouter utilizes a random but fixed hypernetwork (Ha et al., 2017) to generate the router's parameters based on a trainable router embedding. By doing so, HyperRouter can improve the routing policy throughout training while mitigating the representation collapse issues. To demonstrate its effectiveness, we conduct extensive evaluations on various NLP tasks, comparing HyperRouter to several state-of-the-art SMoE routing strategies. Moreover, HyperRouter achieves the same performance threshold with fewer experts (compute) during inference, thereby significantly enhancing the efficiency of deploying LLMs in real-world applications. Fig. 1 shows that HyperRouter consistently outperforms other competitors when the same number of experts is used during inference.

## 2 Related Work

**Existing Routing Mechanism for SMoE.** One of the most important components for training SMoE is the expert routing strategy, which specifies experts to process input tokens. There are two common classes of token-expert assignment algorithms for SMoE training: (1) letting tokens select the top-$k$ experts and (2) letting experts select the top-$k$ tokens. For the first approach, (Fedus et al., 2022b, Switch Transformer), (Lepikhin et al., 2021, GShard), (Zuo et al., 2022, THOR), (Lewis et al., 2021, BASE), (Clark et al., 2022, S-BASE), and SMoE-Dropout (Chen et al., 2023) are representative methods. Meanwhile, Zhou et al. (2022) introduced *expert choice* that enables selecting different experts for each token and demonstrates the potential of the second approach.

**On the Representation Collapse of SMoE.** A major research focus is how to improve the token-expert assignment to avoid the representation collapse issue where: (i) all the inputs are routed to the same expert (Zuo et al., 2022) or (ii) all

experts converge to similar representations (Chi et al., 2022; Chen et al., 2022). Such issues result in poor specialization among experts, parameter redundancy, and limited performance gains. Existing works address the issue (i) by employing various ad hoc-heuristics, e.g., adding Gaussian noise (Shazeer et al., 2017), limiting the maximum number of inputs that can be routed to an expert (Lepikhin et al., 2021), imposing a load balancing loss (Fedus et al., 2022b), using linear assignment (Lewis et al., 2021), or eliminating the necessity for router networks ((Zuo et al., 2022) employed a consistent regularized loss for stochastic expert assignment, (Roller et al., 2021) incorporated deterministic hashing assignments). To resolve the issue (ii), (Chi et al., 2022) proposed an X-MOE layer to improve the routing algorithm via dimension reduction, $\ell_2$ normalization, and gating temperature. Furthermore, (Dai et al., 2022) proposed Stable-MoE with two training stages to reduce the router's fluctuations. The work related to ours the most is SMoE-Dropout (Chen et al., 2023), which is considered a randomly initialized and fixed router to route tokens. Thus, the routing policy is stable during training and deterministic during inference. Our work improves upon SMoE-Dropout by extending the fixed router to a hypernetwork to further improve the routing policy during training while alleviating the collapsing issues.

## 3 Methodology

This section describes SMoE training and details of the proposed HyperRouter method.

### 3.1 SMoE Training

We first describe SMoE training of LLMs, which consists of a router $\mathcal{R}(\cdot)$ with parameter $W_r$ and $N$ expert networks $\{\mathcal{E}_i(\cdot)\}_{i=1}^{N}$ with parameters $W_{e_i}$. We follow the most common implementation (Fedus et al., 2022b) to use a linear network as the router and split the feedforward networks in LLMs into $N$ separated experts.

**Switch Transformer.** Given an input token $\boldsymbol{x}$ with its representation $\boldsymbol{h} \in \mathbb{R}^d$, the SMoE output $\boldsymbol{y}$ is calculated by routing $\boldsymbol{h}$ only to $k$-most suitable experts determined by the router, i.e.,

$$\boldsymbol{y} = \sum_{j=1}^{N} \mathcal{R}(\boldsymbol{h})_j \cdot \mathcal{E}_j(\boldsymbol{h}) \quad \text{and} \quad (1)$$

$$\mathcal{R}(\boldsymbol{h}) = \text{TopK}(\sigma(W_r \times \boldsymbol{h}), k),$$

where the $\text{TopK}(\cdot, k)$ function keeps the largest $k$

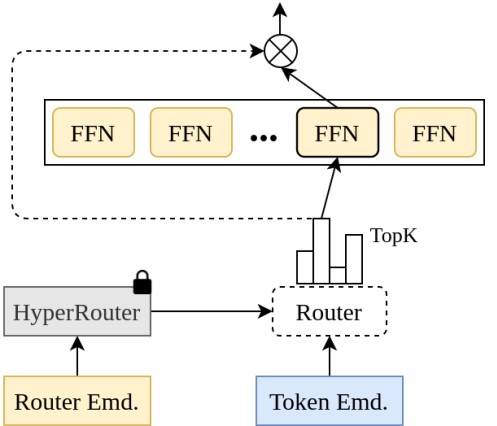

Figure 2: An illustration of our HyperRouter that dynamically generates the router's parameters from a fixed hypernetwork. Yellow modules including the router embedding and experts are trainable while the gray module , the hypernetwork, is frozen. Best viewed in colors.

values of a given vector while setting the remaining values to zero; $\sigma(\cdot)$ is the standard Softmax function. In practice, only a few experts are activated for the current token ($k \ll N$) for efficient training. **SMoE-Dropout.** SMoE-Dropout (Chen et al., 2023) is a state-of-the-art strategy that addresses the representation collapse issues (Sec. 2) by simply fixing a randomly initialized router to improve the token routing consistency. Furthermore, SMoE-Dropout gradually increases the number of experts chosen throughout training, starting with $k = 2$ and ending at $k = N$. During the evaluation, SMoE-Dropout proposes to use *half or all experts* to balance its efficiency and performance.

### 3.2 HyperRouter

We describe our proposed HyperRouter strategy that balances between a random but fixed router and a router optimized during training. HyperRouter employs a **fixed** hypernetwork (Ha et al., 2017) $\mathcal{H}(\cdot)$ to dynamically generate the router's parameters conditioned on a **trainable** router embedding $e$. Specifically, HyperRouter obtains the output $y$ as:

$$y = \sum_{j=1}^{N} \mathcal{R}(h)_j \cdot \mathcal{E}_j(h) \quad \text{and} \quad (2)$$

$\mathcal{R}(h) = \text{TopK}(\sigma(W_r \times h), k)$ and $W_r = \mathcal{H}(e)$, where $e$ is a low dimensional trainable vector associated with the current layer. All parameters are jointly optimized to minimize the task loss. Lastly, like SMoE-Dropout, HyperRouter also gradually increases the number of activated experts through-

out training, starting at $k = 2$ and ending at $k = N$. Fig. 2 visually illustrates our proposed HyperRouter method.

### 3.3 A Deeper Analysis of HyperRouter

We now investigate the representation capabilities of HyperRouter compared to the naive SMoE and SMoE-Dropout. Due to space constraints, we only present the main results here and provide the detailed calculations in Appendix B. First, the naive SMoE jointly trains the router and expert parameters, leading to entanglement and eventually the representation collapse issue (Zuo et al., 2022). SMoE-Dropout proposed to fix the router parameter $W_r$, which we will show to result in a restricted representation. To do so, we need to calculate the Jacobian with respect to $h$, which characterizes how the feature $h$ is involved during training.

**Jacobian of SMoE and SMoE-Dropout.** Let $S_j^{\text{SMOE}}(h) = \sigma(W_r \times h)_j$, $S_j^{\text{Hyper}}(h) = \sigma(\mathcal{H}(e) \times h)_j$, $\mathbb{1}_{ji}$ be the indicator function, and $W_{r,i}$ be the $i$-column of $W_r$, the Jacobians for SMoE and SMoE-Dropout are calculated as:

$$\nabla_h \mathcal{L} = J_1^\top \nabla_y \mathcal{L} + J_2^\top \nabla_y \mathcal{L}$$

$$= J_1^\top \nabla_y \mathcal{L} + \sum_{i=1}^{k} c_i W_{r,i}, \quad \text{where} \quad (3)$$

$$c_i = \sum_{j=1}^{k} S_j^{\text{SMOE}}(h) \left( \mathbb{1}_{ji} - S_i^{\text{SMOE}}(h) \right)$$

$$\times \left( \mathcal{E}_j(h)^\top \nabla_y \mathcal{L} \right).$$

Here, the Jacobian $\nabla_h \mathcal{L}$ is a combination of two terms related to $J_1$ and $J_2$. The $J_1$ component represents how $h$ contributes to the output $y$ and is the same for all methods. The second component related to $J_2$ characterizes learning better experts' representation of $h$ and is the main difference. Since SMoE-Dropout fixes $W_r$, its $\bar{J}_2 = J_2^\top \nabla_y \mathcal{L}$ term is expressed as a linear combination of columns in $W_r$, which lies in a lower dimensional subspace because $W_r \in \mathbb{R}^{k \times d}$ with $k \ll d$ - number of experts is usually smaller than the feature dimension. Thus, SMoE-Dropout alleviates the entanglement between the router and experts at the cost of restricting the experts' representation capabilities.

**Jacobian of HyperRouter.** By a similar calculation, the Jacobian of HyperRouter is shown in equation 4. Given that $e$ is trainable, HyperRouter's Jacobian is not expressed as a simple linear combination in a much lower-dimensional subspace. In contrast,

HyperRouter maps the feature to a subspace whose dimension can be controlled by $e$, which we can easily set to be greater than $k$. Furthermore, by fixing the hypernetwork, the Jacobian does not change freely as in SMoE, which helps alleviate the entanglement. Overall, HyperRouter can alleviate the representation collapse while not sacrificing the representation capabilities.

Jacobian of HyperRouter:

$$\nabla_{\boldsymbol{h}}\mathcal{L} = \boldsymbol{J}_1^\top \nabla_{\boldsymbol{y}}\mathcal{L} + \boldsymbol{J}_2^\top \nabla_{\boldsymbol{y}}\mathcal{L}$$

$$= \boldsymbol{J}_1^\top \nabla_{\boldsymbol{y}}\mathcal{L} + \sum_{i=1}^{k} c_i \mathcal{H}_i(\mathbf{e}), \text{ where} \quad (4)$$

$$\boldsymbol{J}_1 = \sum_{j=1}^{k} S_j^{\text{Hyper}}(\boldsymbol{h}) \nabla_{\boldsymbol{h}}\left(\mathcal{E}_j(\boldsymbol{h})\right),$$

$$c_i = \sum_{j=1}^{k} S_j^{\text{Hyper}}(\boldsymbol{h})\left(\mathbb{1}_{ji} - S_i^{\text{Hyper}}(\boldsymbol{h})\right)$$

$$\times \left(\mathcal{E}_j(\boldsymbol{h})^\top \nabla_{\boldsymbol{y}}\mathcal{L}\right).$$

## 4 Experiments

We conduct experiments to investigate the following hypotheses. First, HyperRouter improves efficiency compared to existing routing strategies, requiring less computation to achieve similar performance thresholds during evaluation. Second, with the same amount of experts used during inference, HyperRouter can achieve better performances compared to other competitors.

### 4.1 Experiment Setting

Most of our experiments follow Chen et al. (2023) and consider a small TransformerXL (Dai et al., 2019) with four layers in our experiments due to the limited resources. We also investigate the scalability by considering larger variants of the TransformerXL with eight or twelve layers. We compare our HyperRouter with several state-of-the-art routing strategies: (i) *SMoE* (Fedus et al., 2022b) - training with trainable routers; (ii) *THOR* (Zhou et al., 2022) - replacing routers with a stochastic experts selection strategy and a consistency regularizer; (iii) *SMoE-Dropout* (Chen et al., 2023) - using a random router and gradually increasing the number of selected experts during training. We also include two traditional baselines: (i) *Dense* - the standard training of transformers where no routing mechanisms are implemented; and (ii) *Dense+Dropout* similar to *Dense* but with Dropout (Srivastava et al., 2014) inserted in the fully connected layers. All

Table 1: Bit-per-character and Perplexity on the enwik8 and WikiText-103 test sets, respectively. Lower is better. $k$ denotes the number of experts chosen during inference. The best results are in **bold**.

| | | | enwik8 | | |
|---|---|---|---|---|---|
| $k$ | Dense | Dense+Dropout | SMoE | SMoE-Dropout | HyperRouter |
| 1 | - | - | 7.20 | 3.02 | **1.48** |
| 2 | - | - | 3.30 | 1.61 | **1.25** |
| 4 | - | - | 1.68 | 1.29 | **1.19** |
| 8 | - | - | 1.32 | 1.19 | **1.17** |
| 16 | 1.25 | **1.16** | 1.27 | **1.16** | 1.16 |
| | | | WikiText-103 | | |
| 1 | - | - | 896.47 | 560.93 | **65.17** |
| 2 | - | - | 146.54 | 89.13 | **37.75** |
| 4 | - | - | 54.69 | 40.00 | **30.45** |
| 8 | - | - | 35.67 | 29.68 | **28.05** |
| 16 | 32.13 | 31.55 | 32.27 | 27.66 | **27.57** |

baselines use the same amount of trainable parameters, while our HyperRouter introduces a neglectable 1024 trainable parameters for the router embeddings.

We consider two training tasks. First, pre-training on the enwik8 (Mahoney, 2011) and WikiText-103 (Merity et al., 2017) datasets, where we follow the same pre-training procedure as (Chen et al., 2023) to train the small and medium models for 400K iterations. We only train the large models for 100K iterations due to resource constraints. Second, finetuning on the SST-2 (Socher et al., 2013), SST-5 (Socher et al., 2013), IMDB (Maas et al., 2011), and BANKING77 (Casanueva et al., 2020) datasets using the model pre-trained on enwik8. Similar to (Chen et al., 2023), we also perform dense finetuning (always choosing all experts) for several epochs. In all experiments, we report the evaluation metrics on the test set when using different numbers of experts for routing methods. More implementation details and additional results are provided in the Appendix.

### 4.2 Pre-training Result

Tab. 1 reports the bit-per-character and perplexity on the enwik8 and WikiText-103 datasets, respectively. First, we observe that SMoE-based training offers improvements over Dense training. Furthermore, adding Dropout can help with Dense training, which achieves comparable performances with SMoE-Dropout, though its improvements diminish on the larger dataset of WikiText-103. The benefit of SMoE training is the efficiency during inference by using fewer experts. Our HyperRouter substantially outperforms both SMoE and SMoE-Dropout on both datasets in this regime. Notably, HyperRouter significantly

Table 2: Bit-per-character on the enwik8 test set using the TransformerXL medium and large models. Lower is better. $k$ denotes the number of experts chosen during inference, and SD denotes the SMoE-Dropout method. The best results are in **bold**.

| | Medium TransformerXL | | | Large TransformerXL | | |
|---|---|---|---|---|---|---|
| k | SMOE | SD | HyperRouter | SMOE | SD | HyperRouter |
| 1 | 2.170 | 3.741 | **1.457** | 1.370 | 1.766 | **1.215** |
| 2 | 1.395 | 1.642 | **1.201** | 1.216 | 1.298 | **1.181** |
| 4 | 1.260 | 1.257 | **1.156** | 1.189 | 1.225 | **1.174** |
| 8 | 1.230 | 1.167 | **1.137** | 1.182 | 1.197 | **1.171** |
| 16 | 1.226 | 1.141 | **1.132** | 1.181 | 1.183 | **1.170** |

Table 3: Transfer performance (Accuracy) on the SST-2, SST-5, IMDB, and BANKING77 datasets. Higher is better. $k$ denotes the number of experts chosen during inference. The best results are in **bold**.

| | SST-2 | SST-5 | IMDB | BANKING77 |
|---|---|---|---|---|
| Dense | 82.18 | 39.72 | 88.30 | 84.54 |
| Dense+Dropout | 81.94 | 39.95 | 88.37 | 84.34 |
| SMoE-Dropout (k=8) | 81.37 | 41.17 | 87.78 | 87.40 |
| HyperRouter (k=8) | **82.87** | **41.26** | **88.43** | **87.66** |
| SMoE (k=16) | 79.98 | 39.45 | 88.16 | 86.52 |
| SMoE-Dropout (k=16) | 81.83 | 41.30 | 87.81 | 87.89 |
| HyperRouter (k=16) | **83.33** | **41.80** | **88.46** | **88.41** |

outperforms SMoE-Dropout when using only one expert, reducing BPC from **3.02** to **1.48** on enwik8, and perplexity from **560.93** to **65.17** on WikiText-103. Overall, with eight experts or less, HyperRouter performs competitively with SMoE-Dropout while only using half of the experts during evaluation.

Tab. 2 reports the BPC on the enwik8 dataset using the medium and large TransformerXL. Notably, we only train the large model 100K iterations due to resource constraints. With larger backbone networks, we observe the gap between our HyperRouter and the baselines becomes more significant, indicating our HyperRouter enjoys good scalability with the model complexity. Lastly, we observe that SMoE-Dropout does not achieve satisfactory results with the large backbone when only trained with 100K and is outperformed by the naive SMoE. This result shows that SMoE-Dropout requires significant resources to achieve good performance. On the other hand, our HyperRouter consistently outperforms both baselines, regardless of the backbone size and number of experts activated.

### 4.3 Finetuning Result

Tab. 3 reports the results of the finetuning experiment on the SST-2, SST-5, IMDB, and BANKING77 datasets. Although we perform dense finetuning,

we also report the results of using only half the experts during evaluation. Overall, we observe consistent accuracy gains from HyperRouter compared to other baselines on all datasets. Notably, on SST-2 and IMDB datasets, HyperRouter with only eight experts could substantially outperform other strategies when using all experts.

### 4.4 Router Analysis

Table 4: Average entropy of the distribution of the routers on the enwik8 dataset. Lower is better.

| Method | Router 1 | Router 2 | Router 3 | Router 4 |
|---|---|---|---|---|
| SMoE | 2.5164 | 2.2126 | 2.2424 | 2.2583 |
| SMoE-Dropout | 2.5849 | 2.1624 | 2.2579 | 2.2690 |
| HyperRouter | **1.4393** | **0.8894** | **1.1269** | **1.3477** |

We now investigate the distributional output from the routers (softmax of router's output) trained with different strategies. Such outputs determine how tokens are assigned to experts. We hypothesize that high-entropy distributions are not preferred since they are closer to being uniform, indicating the router's low confidence in choosing experts for the current token. In the extreme, the router outputs a uniform distribution and assigns tokens to all experts, eventually causing the collapse issue. Therefore, a router with a lower entropy is preferred since it confidently assigns tokens to a few experts, which can improve experts' specialization. To this end, we report the entropy of each router in the small TransformerXL trained on the enwik8 dataset in Tab. 4. The entropy is calculated on all samples in the test set using the model obtained after training. The result clearly shows that HyperRouter achieved much lower entropy than SMoE and SMoE-Dropout. Due to space constraints, we will provide additional visualizations and discussions in Appendix D.

## 5 Conclusion

In this work, we investigated the potentials and limitations of SMoE for training LLMs and proposed HyperRouter to balance between two extremes of SMoE: training and freezing the router. Consequently, HyperRouter can learn better routing policies while alleviating the representation collapse of conventional SMoE training. Experiments on both pre-training and fine-tuning tasks demonstrated promising HyperRouter's capabilities in facilitating efficient, effective training and inference compared to state-of-the-art strategies.

## Limitations

Our work focuses on the efficiency and efficacy of training LLMs using SMoE. Despite the encouraging results, our experiments are conducted only on medium-scale datasets with a small TransformerXL due to computation limitations. Thus, further empirical evaluations are required to validate the scalability of HyperRouter and other SMoE strategies on recent LLMs and larger datasets.

## Ethics Statement

Despite encouraging results, training large-scale LLMs is inevitably costly and requires extensive computational resources, which need to be properly managed. Moreover, our work used data collected on the web, which has been known to suffer from gender and racial biases and requires additional efforts to mitigate its negative impacts. Lastly, our study is a promising step towards facilitating the development of new LLMs, which still requires careful regularization to avoid potential misuses in harmful applications.

## Acknowledgement

This research/project is supported by the National Research Foundation, Singapore, under its AI Singapore Programme (AISG Award No: AISG2-RP-2021-027).

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

The supplementary document of our work is organized as follows. Appendix A provides additional illustrations to compare our HyperRouter with other strategies. Appendix B provides a detailed derivation of the Jacobians. Appendix C provides the detailed experimental settings, additional results, and analysis. Lastly, we conclude this work with a discussion on promising future research venues in Appendix E.

## A Additional Figures

Fig. 3 illustrates the conceptual difference between the traditional SMoE (Fedus et al., 2022b), SMoE-Dropout (Chen et al., 2023), and our HyperRouter. In summary, SMoE uses a trainable router to send tokens to experts. On the other hand, SMoE-Dropout fixed a randomly initialized router and gradually increased the number of experts chosen throughout training. Lastly, HyperRouter improves upon SMoE-Dropout by replacing the router with a fixed hypernetwork and trainable router embedding.

## B Derivation of the Jacobian

This Section details the calculations of the Jacobians presented in Section. 3.3.

The Jacobian of SMoE and SMoE-Dropout is similar to the Jacobian of HyperRouter. Therefore, only the later detailed calculation is shown. Recall

that HyperRouter obtains the output $\boldsymbol{y}$ as follows:
$$\boldsymbol{y} = \sum_{j=1}^{N} \mathcal{R}(\boldsymbol{h})_j \cdot \mathcal{E}_j(\boldsymbol{h}),$$
$$\mathcal{R}(\boldsymbol{h}) = \text{TopK}(\sigma(\mathcal{H}(\mathbf{e}) \times \boldsymbol{h}), k).$$
Here, $\sigma(\cdot)$ is the standard Softmax function given by:
$$\sigma(\mathcal{H}(\boldsymbol{e}) \times \boldsymbol{h})_j = \frac{\exp((\mathcal{H}(\boldsymbol{e}) \times \boldsymbol{h})_j)}{\sum_{i=1}^{N} \exp((\mathcal{H}(\boldsymbol{e}) \times \boldsymbol{h})_i)}$$
$$\equiv S_j^{\text{Hyper}}(\boldsymbol{h}), \ \forall j = 1, \dots, N.$$
Without loss of generality and for simplicity, by rearranging the index of the top $k$-experts, HyperRouter can be rewritten as follows:
$$\boldsymbol{y} = \sum_{j=1}^{k} S_j^{\text{Hyper}}(\boldsymbol{h}) \cdot \mathcal{E}_j(\boldsymbol{h}), \ \text{where}$$
$$S_j^{\text{Hyper}}(\boldsymbol{h}) = \frac{\exp((\mathcal{H}(\boldsymbol{e}) \times \boldsymbol{h})_j)}{\sum_{i=1}^{k} \exp((\mathcal{H}(\boldsymbol{e}) \times \boldsymbol{h})_i)},$$
$$\forall j = 1, \dots, k.$$
The Jacobian matrix $\boldsymbol{J}$ of the output $\boldsymbol{y}$ with respect to $\boldsymbol{h}$ can be decomposed into two terms as follows:
$$\nabla_{\boldsymbol{h}} \boldsymbol{y} = \nabla_{\boldsymbol{h}} \left( \sum_{j=1}^{k} S_j^{\text{Hyper}}(\boldsymbol{h}) \cdot \mathcal{E}_j(\boldsymbol{h}) \right)$$
$$= \sum_{j=1}^{k} S_j^{\text{Hyper}}(\boldsymbol{h}) \nabla_{\boldsymbol{h}} (\mathcal{E}_j(\boldsymbol{h}))$$
$$+ \sum_{j=1}^{k} \nabla_{\boldsymbol{h}} \left( S_j^{\text{Hyper}}(\boldsymbol{h}) \right) \mathcal{E}_j(\boldsymbol{h})$$
$$= \sum_{j=1}^{k} S_j^{\text{Hyper}}(\boldsymbol{h}) \nabla_{\boldsymbol{h}} (\mathcal{E}_j(\boldsymbol{h}))$$
$$+ \sum_{j=1}^{k} \sum_{i=1}^{k} S_j^{\text{Hyper}}(\boldsymbol{h}) \left( \mathbb{1}_{ji} - S_i^{\text{Hyper}}(\boldsymbol{h}) \right)$$
$$\times \mathcal{E}_j(\boldsymbol{h}) \mathcal{H}_i(\mathbf{e})^{\top} \equiv \boldsymbol{J}_1 + \boldsymbol{J}_2 \quad (5)$$
(since $\mathbf{e}$ is independent of $\boldsymbol{h}$).

Note that the last equation in (5) is obtained by using the chain rule, the logarithmic derivative, and the inner product as follows:
$$\frac{\partial s_j}{\partial z_i} = s_j \cdot \frac{\partial}{\partial z_i} \log(s_j) = s_j \cdot (\mathbb{1}_{ji} - s_i), \ \text{where}$$
$$s_j = \frac{\exp(z_j)}{\sum_{i=1}^{k} \exp(z_i)}, \ \forall j = 1, \dots, k.$$
It is worth mentioning that the first term $\boldsymbol{J}_1$ means to produce a better token representation given the current activation router $S_j^{\text{Hyper}}(\boldsymbol{h})$, while the second term $\boldsymbol{J}_2$ represents learning a better gating function for the appropriate activation score router $S_j^{\text{Hyper}}(\boldsymbol{h})$. After the back-propagation, the gradi-

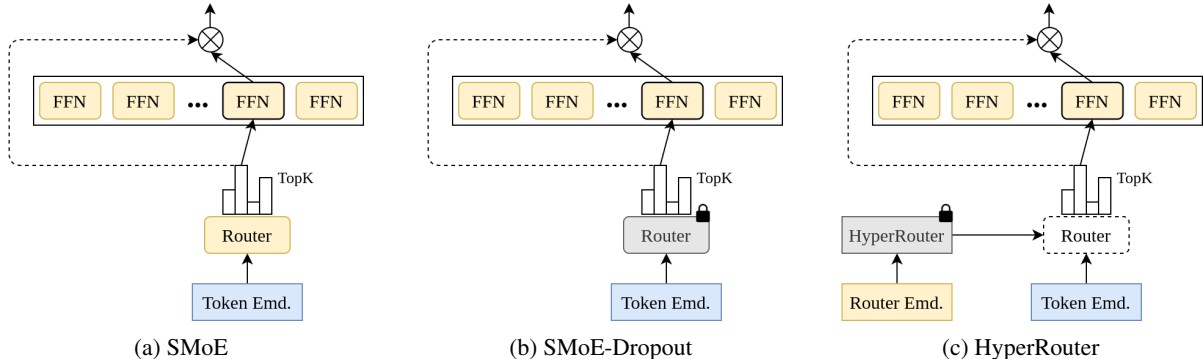

(a) SMoE          (b) SMoE-Dropout          (c) HyperRouter

Figure 3: An illustrative comparison among SMoE, SMoE-Dropout, and our HyperRouter . The input token is a representation vector (could be output from the previous layer). Yellow modules are trainable. Gray modules are frozen, indicated by a lock symbol (🔒). Best viewed in colors.

ent of the loss function $\mathcal{L}$ is obtained from the two paths mentioned above and is written as follows

$$\nabla_{\boldsymbol{h}}\mathcal{L} = \nabla_{\boldsymbol{y}}\mathcal{L} \times \nabla_{\boldsymbol{h}}\boldsymbol{y} = \nabla_{\boldsymbol{h}}\boldsymbol{y}^{\top} \times \nabla_{\boldsymbol{y}}\mathcal{L}$$
$$= (\boldsymbol{J}_1 + \boldsymbol{J}_2)^{\top} \times \nabla_{\boldsymbol{y}}\mathcal{L} \quad \text{(using (5))}$$
$$= \boldsymbol{J}_1^{\top}\nabla_{\boldsymbol{y}}\mathcal{L} + \boldsymbol{J}_2^{\top}\nabla_{\boldsymbol{y}}\mathcal{L}. \quad (6)$$

Finally, by expanding the second term as follows, we obtain the desired result:

$$\boldsymbol{J}_2^{\top}\nabla_{\boldsymbol{y}}\mathcal{L} = \sum_{i=1}^{k}\sum_{j=1}^{k} S_j^{\text{Hyper}}(\boldsymbol{h})\left(\mathbb{1}_{ji} - S_i^{\text{Hyper}}(\boldsymbol{h})\right)$$
$$\times \left(\mathcal{E}_j(\boldsymbol{h})^{\top}\nabla_{\boldsymbol{y}}\mathcal{L}\right)\mathcal{H}_i(\mathbf{e})$$
$$= \sum_{i=1}^{k} c_i \mathcal{H}_i(\mathbf{e}), \text{ where} \quad (7)$$
$$c_i = \sum_{j=1}^{k} S_j^{\text{Hyper}}(\boldsymbol{h})\left(\mathbb{1}_{ji} - S_i^{\text{Hyper}}(\boldsymbol{h})\right)$$
$$\times \left(\mathcal{E}_j(\boldsymbol{h})^{\top}\nabla_{\boldsymbol{y}}\mathcal{L}\right).$$

## C   Additional Experiments

This section provides the implementation details of our experiments in Sec. 4.

### C.1   General Setting

Our experiments are conducted based on the publicly available SMoE-Dropout (Chen et al., 2023) implementation[1]. Moreover, the pre-training experiments were conducted on a single A40 or A100 GPU, while the finetuning experiments were conducted on a single GeForce RTX 3090 GPU. We also emphasize that parallel training on multiple GPUs might yield different results.

---

[1] https://github.com/VITA-Group/Random-MoE-as-Dropout

**Model architecture.** The small TransformerXL variant (Chen et al., 2023) consists of 4 Transformer decoder layers with an input dimension of 256. Each layer consists of a self-attention layer with 8 attention heads, followed by a feedforward network with an inner dimension of 512. The dropout ratio is kept at 0.1. We split the feedforward network into 16 experts with the same dimension. For the HyperRouter , we initialize the embeddings with size 256 and employ a 2-layer perceptron with an inner dimension of 256 as the hypernetwork. The ReLU function is used as the activation function for the hypernetwork. Totally our HyperRouter introduces an additional 1024 trainable parameters in the TransformerXL model. It is worth noting that the parameter overhead is fixed as we scale up to larger transformer models. For the medium and large variants, we scale the model to eight and twelves layers, respectively.

### C.2   Pre-training Experiments

Tab. A1 provides the implementation details for pre-training our TransformerXL small and medium on enwik8 and WikiText-103. The TransformerXL large network was trained in the same maner, but only for 100K iterations.

Table A1: Implementation details for pre-training experimentson enwik8 and WikiText-103 datasets.

| Dataset | Input length | Batch size | Optimizer | Lr | # Iterations |
|---|---|---|---|---|---|
| enwik8 | 512 | 22 | Adam | 2.5e-4 | 400000 |
| WikiText-103 | 512 | 22 | Adam | 2.5e-4 | 400000 |

## C.3 Finetuning Experiments

To conduct finetuning experiments, we use the same model architecture as in pre-training. Tab. A2 provides the implementation details used for finetuning experiments on four different datasets.

Table A2: Implementation details for finetuning experiments on four different datasets.

| Dataset | Input length | Batch size | Optimizer | Lr | # Epochs |
|---------|-------------|-----------|-----------|-----|---------|
| SST-2 | 512 | 16 | Adam | 1e-4 | 3 |
| SST-5 | 512 | 16 | Adam | 1e-4 | 3 |
| IMDB | 512 | 4 | Adam | 1e-4 | 3 |
| BANKING77 | 512 | 16 | Adam | 1e-4 | 15 |

## C.4 Inference Efficiency

We discuss an implementation detail that makes our HyperRouter have the same efficiency during inference as SMoE and SMoE-Dropout. Particularly, before performing inference on any input samples, we use the hypernetwork to generate the router for each layer and then store them. During inference, HyperRouter does not need to re-generate routers. Therefore, HyperRouter enjoys the same inference complexity as SMoE and SMoE-Dropout because their routers have the same dimensionality. Tab. A3 reports the number of FLOPs of different methods during inference on the enwik8 dataset according to a different number of experts chosen ($k$). Note that the FLOPs do not scale linearly with $k$ since SMoE training is only applied to the fully connected layers, while other layers (multi-head self-attention, layer normalization, etc.) remain the same across baselines. In the extreme, using only one expert only incurs 45% of the FLOPs compared to using all 16 experts.

Table A3: Inference FLOPS ($10^{10}$) on the enwik8 dataset, $k$ denotes the number of experts used during inference.

| $k$ | Dense | Dense+Dropout | THOR | SMoE | SMoE-Dropout | HyperRouter |
|-----|-------|--------------|------|------|-------------|-------------|
| 1 | - | - | - | 3.47 | 3.47 | 3.47 |
| 2 | - | - | - | 4.00 | 4.00 | 4.00 |
| 4 | - | - | - | 4.54 | 4.54 | 4.54 |
| 8 | - | - | - | 5.61 | 5.61 | 5.61 |
| 16 | 7.76 | 7.76 | 7.66 | 7.66 | 7.66 | 7.66 |

## C.5 Parameter Comparison

Tab. A4 provides the number of parameters in different components of SMoE-Dropout and HyperRouter . There are three categories: (i) trainable parameters which are the transformer

Table A4: Number of parameters in different components of SMoE-Dropout and HyperRouter during training. Blue parameters are trainable, red parameters are frozen, and underline parameter are dynamically generated in each iteration.

| Method | Router Emb. | Router | HyperNetwork | Transformer |
|--------|-------------|--------|-------------|-------------|
| SMoE-Dropout | − | 16.448 | − | 19, 505.350 |
| HyperRouter | 1.024 | 16.448 | 1, 112.576 | 19, 505.350 |

backbone and the router embeddings; (ii) frozen parameters which are the routers SMoE-Dropout and hypernetworks in HyperRouter ; (iii) dynamic parameters that are dynamically generated in each iteration, such as the routers in HyperRouter . Overall, the additional trainable parameters in HyperRouter is neglectable. Moreover, the number of frozen parameters (hypernetworks) is quite small compared to the transformer backbone (5.7%). Investigating into sharing the hypernetworks across layers or generating the routers coordinately (Pham et al., 2022) can further reduce this cost while improving the performance, which we will leave for the future work.

## D    Routing Visualization

Table A5: Average entropy of the distribution of the routers' output on the enwik8 dataset. Lower is better.

| Method | Router 1 | Router 2 | Router 3 | Router 4 |
|--------|----------|----------|----------|----------|
| SMoE | 2.5164 | 2.2126 | 2.2424 | 2.2583 |
|  | ± 0.17 | ± 0.31 | ± 0.30 | ± 0.25 |
| SMoE-Dropout | 2.5849 | 2.1624 | 2.2579 | 2.2690 |
|  | ± 0.11 | ± 0.37 | ± 0.28 | ± 0.27 |
| HyperRouter | **1.4393** | **0.8894** | **1.1269** | **1.3477** |
|  | ± 0.42 | ± 0.46 | ± 0.46 | ± 0.50 |

This Section provides the full details of the routers' entropy and visualizes its distributional output. This is supplementary to Section 4.4. Tab. A5 reports the mean and standard deviations of the entropy at each router. This table is the full version of Tab. A5. We can see that all methods have rather low standard deviation, indicating that the differences are significant.

We also provide an illustrative example of the routers' outputs using a randomly picked sample on the test set in Fig. 4. Here we can clearly see that the policies from SMoE and SMoE-Dropout are much closer to uniform while HyperRouter 's policies are much sharper and have lower entropy. We emphasize that this example is not cherry-picked since we already calculated the averaged entropy

on all samples in Tab. A5. Overall, this result shows insights into how HyperRouter can perform better than other state-of-the-art SMoE strategies.

## E    Future Work

Our HyperRouter opens several promising venues for future research. Particularly, we believe that investigating two HyperRouter components: (i) fixed hypernetwork, and (ii) trainable embedding can yield further performance and efficiency gains. Potentials directions include incorporating regularization such as $\ell_2$-penalty or dropout (Peng et al., 2015) or using better hypernetwork initialization techniques (Chang et al., 2020). Furthermore, the current implementation uses a hypernetwork for each transformer layer, and it generates all parameters of the router. Sharing hypernetworks among layers or generating the router coordinate-wise can offer knowledge sharing (Yin et al., 2021; Pham et al., 2022), which can further improve the result.

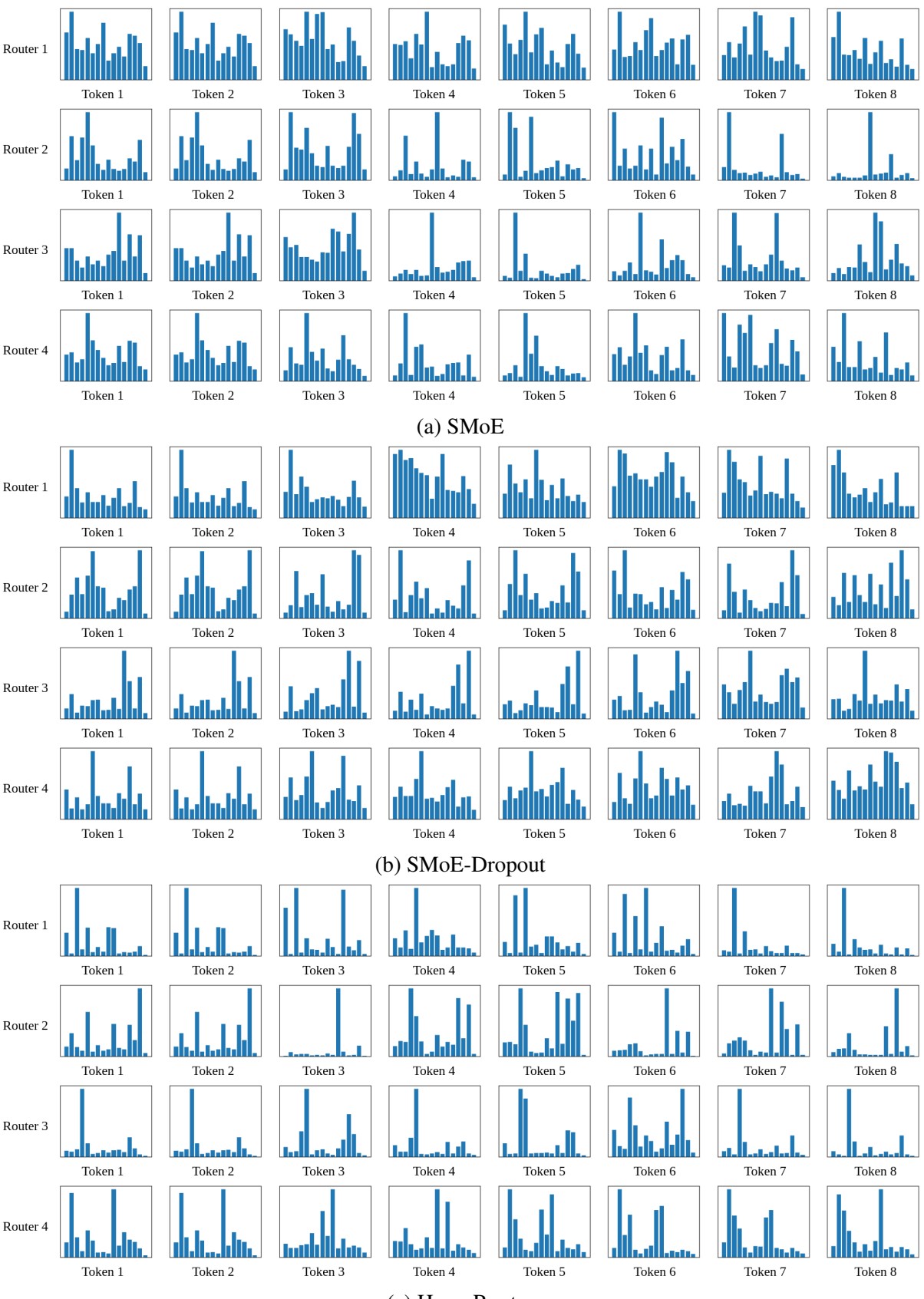

Figure 4: Visualization of the distribution of the routers' output.