# OpenReview forum: "HyperRouter: Towards Efficient Training and Inference of Sparse Mixture of Experts"
_EMNLP/2023/Conference — EMNLP 2023 Main_

### Official Review · Reviewer_wUFz · 2023-07-29

**Typos Grammar Style And Presentation Improvements:** 1. Line 108-110. "enable selecting di…
**Soundness:** 3

**Excitement:**

4: Strong: This paper deepens the understanding of some phenomenon or lowers the barriers to an existing research direction.

**Paper Topic And Main Contributions:**

This paper proposes HyperRouter to address the tradeoff between fixed and trainable routers in SMoE training.

SMoE trains the router and suffers from representation collapse. SMoE-Droput effectively mitigates the collapse, but its random routing policy might be sub-optimal and its inference requires half or all of the experts, leading to limited efficiency gains during inference. This work proposes using a random but fixed hyper-network to generate the router's parameters based on a trainable router embedding.

This work shows empirically that HyperRouter achieves the same performance with fewer experts (compute) during inference, significantly enhancing inference efficiency. It also consistently outperforms competitors when the same number of experts is used during inference.

**Questions For The Authors:**

A. What are the empirical results on large and medium scale transformers (LLMs)? If there is indeed resource limitation, could you provide some hypothesis and minimal experiment results to validate your hypothesis?

B. In section 3.2,  it's not clear why the decoupling of HyperRouter's embeddings from the token representation will improve the routing policy. Is it from some property of hypernetwork or something else? There is not enough context on hypernetwork and why the proposed approach will lead to improvements. It might not be obvious to readers.

[Update] The authors addressed the two questions with proper empirical evidence and explanation in the rebuttal.

**Reasons To Accept:**

The results based on a small scale Transformer-XL on both pretraining and finetuning tasks are very convincing. It shows for pretraining tasks and during inference, HyperRouter achieves the same performance as competitors with half the number of experts, warrant its advantage for LLM deployment. It also shows for finetuning tasks, there is consistent accuracy gains over competitors on all datasets.

The additional experiment results about the inference FLOPs, parameter comparison, and entropy analysis and routing visualization in the appendix add more strength to the analysis. They provide extra angles to help understand why the proposed approach is advantageous.

**Reasons To Reject:**

All experiment results are based on a small-scale transformer-XL. Based on the configuration in the paper, the model has about 1.8M parameters. The SMoE-Droput paper reported results on BERT base (110M parameters), RoBERTa base (125M), and Transformer-XL (18M).  So the reader of this paper may not have a clear picture of what the performance gains and inference advantages are on large and medium scale transformers (LLMs). Will the conclusions generalize to larger-scale LLMs or the gain will diminish as it scales? We need more empirical evidence.

[Update] The authors addressed the two questions with proper empirical evidence and explanation in the rebuttal.

**Reproducibility:**

4: Could mostly reproduce the results, but there may be some variation because of sample variance or minor variations in their interpretation of the protocol or method.

**Reviewer Confidence:**

4: Quite sure. I tried to check the important points carefully. It's unlikely, though conceivable, that I missed something that should affect my ratings.

---

> ### Author Rebuttal · Authors · 2023-08-28
>
> We refer the Reviewer to our general responses for the detailed analysis of the performance gain from HyperRouter. We also conducted a larger scale experiment with TransformerXL-Medium and TransformerXL-Large. The result indicates that the gap between HyperRouter is consistent, and even becomes larger, as we increase the model complexity. This result is also consistent with our analysis (please see "Explaining the Performance Gains from HyperRouter (comments from Reviewer EK3o and wUFz)"), which does show that the expert's representation capabilities in SMoE-Dropout is restricted to a low dimensional subspace.
>
> > Confusion regarding Line 108-110
>
> We apolgize for the confusion in L108-110. There are 2 categories: (i) each token chooses k experts; and (ii) each expert chooses k tokens. SMoE, SMoE-Dropout, and our HyperRouter fall into the first category while THOR (Zhou et al., 2022) falls into the second category.
>
> In the revision, we will revise this section to discuss the literature more clearly.
>
> > The section B.4 (inference FLOPS) and C (routing visualization) are very useful
>
> Thank you for the suggestion. In the revision, with an additional page, we will make sure in include and discuss this result in the main paper.

---

### Official Review · Reviewer_1481 · 2023-08-02

**Soundness:** 3

**Excitement:**

4: Strong: This paper deepens the understanding of some phenomenon or lowers the barriers to an existing research direction.

**Paper Topic And Main Contributions:**

The paper presents HyperRouter, a dynamic routing approach that generates the router's parameters through a fixed hypernetwork and trainable embeddings. This method strikes a balance between training the routers and freezing them, resulting in improved routing policies and alleviating the representation collapse issue encountered in Sparse Mixture-of-Experts (SMoE). The authors claim that extensive experiments across various NLP tasks demonstrate the superior performance of HyperRouter compared to existing routing methods for SMoE while using fewer experts during inference.




**Reasons To Accept:**

- HyperRouter introduces a novel approach that balances fixed and trainable routers in SMoE training, enhancing the routing policy and mitigating representation collapse issues, resulting in more efficient and effective large language model training.

- The authors conduct experiments on both pre-training and fine-tuning for various datasets and note that HyperRouter:
a) Outperforms both SMoE and SMoE-Dropout (in the pre-training regime)
b) Significantly outperforms SMoE-Dropout when using only one expert (in the pre-training regime)
c) Performs competitively with SMoE275 Dropout while only using half of the experts (in the pre-training regime)
d) In the fine-tuning regime- it outperforms all the other strategies on SST-2 and IMDB datasets with only 8 experts



**Reasons To Reject:**

- The experiments in the paper have a narrow scope since the authors use only one kind of transformer architecture. The results are are not very conclusive for future research to build on.

- I'm not sure if utilizing just 4 layers (out of 18) from TransformerXL would accurately predict the model's behavior. The authors might have opted for a smaller model for directly translating the results.

**Reproducibility:**

4: Could mostly reproduce the results, but there may be some variation because of sample variance or minor variations in their interpretation of the protocol or method.

**Reviewer Confidence:**

4: Quite sure. I tried to check the important points carefully. It's unlikely, though conceivable, that I missed something that should affect my ratings.

---

> ### Author Rebuttal · Authors · 2023-08-28
>
> We refer the Reviewer to our General Response "Large Scale Experiments" for the results and detailed discussions on the large scale models. In summary, we observe that the gap between our HyperRouter and SMoE-Dropout becomes larger as we increase the model complexity, which is promising for scaling our method to larger scales LLMs.

---

### Official Review · Reviewer_EK3o · 2023-08-10

**Soundness:** 2

**Excitement:**

3: Ambivalent: It has merits (e.g., it reports state-of-the-art results, the idea is nice), but there are key weaknesses (e.g., it describes incremental work), and it can significantly benefit from another round of revision. However, I won't object to accepting it if my co-reviewers champion it.

**Paper Topic And Main Contributions:**

In this paper, the authors introduce a new approach that features dynamic router parameter generation in sparse MoE. The authors argue that the existing approach has limitations on the sub-optimal random routing policy and limited efficiency gains for inference. To mitigate these limitations, this paper presents HyperRouter, which generates the router's parameters based on a trainable router embedding. The evaluation shows that HyperRouter efficiently improves the inference performance.

**Reasons To Accept:**

1. The authors propose an interesting approach that facilitates dynamic router parameter generation.

2. The evaluation demonstrates the potential and performance improvement of the proposed approach.

**Reasons To Reject:**

1. The paper needs a comprehensive analysis of sparse MoE, including the communication overhead (all to all). Currently, it's not clear where the performance gain comes from, basically, different number of experts incurs different communication overhead.

2. The evaluation needs experiments on distributed deployment and a larger model.

3. For the arguments that the existing approach has two key limitations, the authors should present key experiment results for demonstration.

**Reproducibility:**

3: Could reproduce the results with some difficulty. The settings of parameters are underspecified or subjectively determined; the training/evaluation data are not widely available.

**Reviewer Confidence:**

3: Pretty sure, but there's a chance I missed something. Although I have a good feel for this area in general, I did not carefully check the paper's details, e.g., the math, experimental design, or novelty.

---

> ### Author Rebuttal · Authors · 2023-08-28
>
> We refer the Reviewer to our general comments for the intuition of our method and the large scale experiment results.
>
> > The paper needs a comprehensive analysis of sparse MoE, including the communication overhead (all to all).
>
> Thank you for the interesting suggestion. However, we would appreciate if the Reviewer could elaborate more on this concern regarding all-to-all communication overhead.
>
> For this work, we focus on designing a more effective routing policy by addressing the weaknesses of naive SMoE and SMoE-Dropout, which are two extremes of SMoE training. Our HyperRouter incurs neglectable additional trainable parameters (layer embeddings) and has the same training/inference complexities as SMoE-Dropout (as we showed in Appendix B.4 and B.5).
>
> > The evaluation needs experiments on distributed deployment and a larger model
>
> We appreciate the Reviewer's feedback. The main focus of our work is to propose a simple and effective router for SMoE-based training of LLM. We have shown encouraging results of HyperRouter compared to the standard SMoE training and a state-of-the-art strategy, SMoE-Dropout. Unfortunately, we are not able to conduct large scale distributed training due to our limited computation resources. Even so, we tried our best to conduct larger scale experiments with TransformerXL-Medium and TransformerXL-Large in this rebuttal. The results consistently show that our HyperRouter can offer improvements to the baselines across different network sizes and number of experts used during evaluation. Thus, we firmly believe that any advanced in distributed training of SMoE can be easily adopted to HyperRouter with similar performance gains.
> However, a comprehensive evaluation of large-scale distributed training is beyond the scope of this study and we will leave this for the future work.
>
> > For the arguments that the existing approach has two key limitations, the authors should present key experiment results for demonstration.
>
> Our work argue two limitations of SMOE-Dropout: (i) the routing policy from the randomly initialized gating might be sub-optimal; and (ii) SMoE-Dropout requires more experts to achieve good performances.
>
> Regarding (i), it is clear that a better routing policy will result in a better final performance, giving the same network architecture. This result is supported by the experiments throughout our study, e.g. Table 1, 2 and Figure 1 where HyperRouter consistently outperforms SMoE-Dropout given the same network architecture. Moreover, in Appendix C, we also visualize the routing policies and report their entropy from different methods. We can clearly see that HyperRouter's routing policy has much lower entropy, indicating the router can confidently assign tokens to experts. For a more detailed discussion, please refer to Appendix C.
>
> The arguement (ii) is supported by Figure 1, Table 1, and 2 where our HyperRouter can achieve similar performances with SMoE-Dropout when using only half of the experts during evaluation.

---

### Meta-Review · Area_Chair_tYVD · 2023-09-18

**Recommendation:** 4

**Metareview:**

The paper presents a dynamic routing approach for Sparse Mixture-of-Experts (SMoE) training and inference. This method aims to mitigate the limitations of existing SMoE approaches by generating the router's parameters through a combination of a fixed hypernetwork and trainable embeddings.

Generally, the reviewers appreciated the novelty of the proposed approach and that experimental results were given for both pre-training and fine-tuning settings. The results indicate that the approach appears to outperform existing routing strategies in the SMoE regime. Some concerns were raised by reviewers around the size of models evaluated, to which the authors provided additional results. These results indicate that the proposed approach also works on larger models (at least in the architectural family). The paper would be even stronger if additional Transformer architectures were explored.

There was also concerns related to communication overhead/amount of computation of the approach raised by one reviewer. It is not clear to me why this approach would suffer additional communication overhead compared to relevant existing SMoE approaches. The authors state clearly in the rebuttal that it is a fair comparison in terms of number of experts used across the different approaches.

Finally, the authors also did a good job in the rebuttal of providing additional analysis to argue why the representational power of HyperRouter could be considered better than prior work using dropout.

---

### Decision · Program_Chairs · 2023-10-07

**Decision:**

Accept-Main

**Comment:**

The paper presents a dynamic routing approach for Sparse Mixture-of-Experts (SMoE) training and inference. This method aims to mitigate the limitations of existing SMoE approaches by generating the router's parameters through a combination of a fixed hypernetwork and trainable embeddings.

Generally, the reviewers appreciated the novelty of the proposed approach and that experimental results were given for both pre-training and fine-tuning settings. The results indicate that the approach appears to outperform existing routing strategies in the SMoE regime. Some concerns were raised by reviewers around the size of models evaluated, to which the authors provided additional results. These results indicate that the proposed approach also works on larger models (at least in the architectural family). The paper would be even stronger if additional Transformer architectures were explored.

There was also concerns related to communication overhead/amount of computation of the approach raised by one reviewer. It is not clear to me why this approach would suffer additional communication overhead compared to relevant existing SMoE approaches. The authors state clearly in the rebuttal that it is a fair comparison in terms of number of experts used across the different approaches.

Finally, the authors also did a good job in the rebuttal of providing additional analysis to argue why the representational power of HyperRouter could be considered better than prior work using dropout.